# RUNX Family as a Promising Biomarker and a Therapeutic Target in Bone Cancers: A Review on Its Molecular Mechanism(s) behind Tumorigenesis

**DOI:** 10.3390/cancers15123247

**Published:** 2023-06-19

**Authors:** Selvaraj Vimalraj, Saravanan Sekaran

**Affiliations:** Department of Prosthodontics, Saveetha Dental College and Hospital, Saveetha Institute of Medical and Technical Sciences (SIMATS), Saveetha University, Chennai 600077, Tamil Nadu, India

**Keywords:** osteosarcoma, RUNX2, tumorigenesis, bone

## Abstract

**Simple Summary:**

The transcription factor RUNX family plays a crucial role in the formation of osteoblasts from bone marrow mesenchymal stem cells, contributing to bone development. However, emerging evidence suggests its involvement in tumor biology and cancer progression. In particular, the RUNX family has been associated with osteosarcoma by regulating factors related to tumorigenicity. As a vital molecule within the regulatory network of cancers, RUNX influences various upstream signaling pathways and downstream target molecules. Understanding the precise mechanisms underlying the development and prognosis of malignant tumors is essential. RUNX has been recognized as a potential therapeutic target for bone cancer, and further research into its role may lead to the development of new medications and improved clinical outcomes.

**Abstract:**

The transcription factor runt-related protein (RUNX) family is the major transcription factor responsible for the formation of osteoblasts from bone marrow mesenchymal stem cells, which are involved in bone formation. Accumulating evidence implicates the RUNX family for its role in tumor biology and cancer progression. The RUNX family has been linked to osteosarcoma via its regulation of many tumorigenicity-related factors. In the regulatory network of cancers, with numerous upstream signaling pathways and its potential target molecules downstream, RUNX is a vital molecule. Hence, a pressing need exists to understand the precise process underpinning the occurrence and prognosis of several malignant tumors. Until recently, RUNX has been regarded as one of the therapeutic targets for bone cancer. Therefore, in this review, we have provided insights into various molecular mechanisms behind the tumorigenic role of RUNX in various important cancers. RUNX is anticipated to grow into a novel therapeutic target with the in-depth study of RUNX family-related regulatory processes, aid in the creation of new medications, and enhance clinical efficacy.

## 1. Introduction

The age distribution of osteosarcoma (OS), the most frequent kind of primary malignant bone sarcoma, is bimodal. The biggest risk group is children and teenagers (median age 18), followed by individuals over the age of 60. There are between three and five cases of OS per one million people each year [1]. The long bones (femur, tibia, and humerus) are the most common sites for bone tumor development, followed by the head, jaw, and pelvis. This is because these regions are so close to the bone’s development plate, also known as the metaphysis. The osteoid matrix, produced by converted osteoblastic cells, is a defining feature of OSs. The origin of the malignant cell, however, is still a mystery. Mesenchymal stem cells (MSCs) are pre-osteoblastic cells that are thought to be the starting point for osteogenesis imperfecta (OS) [2,3,4]. Osteosarcoma (OS) patients have been receiving neo-adjuvant treatment, followed by adjuvant therapy after surgery, since the late 1970s when chemotherapies were first used to treat OS. In the French OS2006/sarcome-09 research [3], young people under the age of 25 who have cancer are treated with a combination of chemotherapies including high-dose methotrexate ifosfamide and etoposide. Doxorubicin, cisplatin, and ifosfamide, with or without high-dose methotrexate, are among the other combinations used in cancer treatment regimens [4]. The five-year survival rate for kids and young adults whose cancer has been contained thanks to these treatments is 78%, but it remains at just 20% for those who have metastases at diagnosis or recurrence. There have been no major gains in survival rates over the last 40 years [4] for patients with or without metastases. There are several organizations working to better understand and treat OS all around the globe.

Runt-related transcription factor 2 (RUNX2) is a member of the RUNX family of transcription factors, which also includes RUNX1 and RUNX3, and is documented as playing an essential role in the regulation and control of transcription during osteogenesis, prostate development, and skeletal growth [3,4,5]. Via its regulation of various signaling cascades (e.g., MAPK and NK-B pathway macrophage reprogramming), RUNX2 has been connected with various physiological processes including endothelial cell migration, osteogenic differentiation, immunomodulation, chondrocyte hypertrophy, and vascular invasion [6,7]. Atherosclerosis [8], skeletal dysplasia (including cleidocranial dysplasia), and cancer [7] have all been linked to abnormal RUNX2 expression or activity. RUNX2 is involved in a wide variety of cancers, including gastric cancer, breast renal cell carcinoma, and osteosarcoma [7]. As an example, a recent study has shown [8] that the RUNX2 activation of MMP1 significantly correlates with a poor prognosis by promoting the aggressiveness and chemoresistance of TNBC cells. Notably, RUNX2 has been linked to both CSCs and bone cancer growth in prior research [8]. Together, these findings point to a role for RUNX2 in BCSCs, in the diagnosis of bone cancer, and in treatment resistance, all of which provide credence to the hypothesis that RUNX2 might serve as a therapeutic target for anticancer drugs in the future. The functions of RUNX2 in the development and treatment of bone cancer were the primary foci of this review. Recent developments in the treatment of bone cancer, from RUNX2-based therapies to the establishment of drug resistance, were also addressed in depth. The promise of RUNX2 as a diagnostic and therapeutic target for bone cancer has grown as more has been learned about it.

## 2. The RUNX Family

Gene expression regulation by the RUNX family of transcription factors has significant effects on both embryonic development and cell differentiation. RUNX transcription factors have been shown to be essential for the development of many different kinds of hematopoietic cells. RUNX1 and RUNX3 regulate blood cell synthesis during the early phases of cell lineage determination, whereas RUNX2 is essential for bone growth at later stages. One of the first genes to be associated with AML was RUNX1. Mutations in this gene are common in lymphoblastic and myeloid leukemia [9]. Hereditary platelet abnormalities have been linked to RUNX1 deficiency. Myeloid leukemia tends to run in families, and its symptoms include a decrease in white blood cell counts (thrombocytopenia) and the number of B and T lymphocytes [10]. RUNX2 is essential for the growth of bone and cartilage, and it promotes the spread of tumors to bones [11]. RUNX2 has been linked to many different malignancies, including melanoma, thyroid, and liver tumors [12,13]. According to Guan et al. [14], circ RNA_102272 enhanced the cisplatin resistance of hepatocellular carcinoma via blocking miR-326’s influence on RUNX2. In particular, it is reported that RUNX2 could significantly induce glycolysis and oxidative phosphorylation, resulting in increased mitochondrial activity, suggesting the promoting effect of RUNX2 in accelerating tumor cell metabolism and, in particular, in favoring the invasion and migration of leukemic cells via mediating the interaction between glycolysis and mitochondrial respiration. In addition to regulating cell proliferation, RUNX2 promotes the self-renewal of gastric cancer cells and aids in tumor invasion and metastasis [15]. By binding to miR-15 family members, repressing LDL receptor-related protein 6 expression, and boosting b-catenin signalling, or by binding to SFPQ (splicing factor proline and glutamine-rich) and dissociating the SFPQ/polypyrimidine tract binding protein 2 dimer, Ji et al. [16] found that metastatic lung adenocarcinoma transcript 1 could regulate the transcription and translation of RUNX2. Altogether, RUNX2 appears promising as a biomarker for the early diagnosis of several malignancies and as a therapeutic target for the treatment of patients. RUNX3 is essential for the growth and metastasis of many different forms of human cancer. Zhang et al. [17] found that RUNX3 suppresses the development and metastasis of colorectal cancer. Liu et al. [18] observed that RUNX3 suppressed glutamine metabolism in gastric cancer patients. Researchers have found a link between the NO•/RUNX3/kynurenine metabolic axis [19] and the lethality of pancreatic cancer. Therefore, the abnormal expression of any member of the RUNX family has the potential to induce a wide range of disorders since these members are engaged in typical physiological functions. An overview of the Runx family, including a breakdown of where somatic mutations are located in each of the three human RUNX genes, is shown in Figure 1.

## 3. RUNX2’s Physiological and Pathological Functions

RUNX2 is one of the transcription factors that make up the RUNX family. Osteosarcoma, osteoarthritis, prostatic carcinoma, breast cancer, and gastric cancer [20,21,22,23,24,25] have all been linked to RUNX2, which also plays a role in the differentiation and maturation of osteoblasts and chondrocytes [21,23]. Multiple cell types, such as osteoblasts, chondrocytes, and MSCs, rely on RUNX2 for normal functioning [26]. In addition to its importance in breast development [27], RUNX2 interacts favorably with TWIST1 to control the destiny of cells originating from the cranial neural crest and direct the creation of the face muscles [28]. Single-cell transcriptome analysis, cell sorting, and lineage tracing all reveal that RUNX2 is essential for bone production and osteoblast differentiation. For example, Shu et al. [29] created a dual-recombinase fate-mapping system to monitor the geographical and temporal transition of skeletal progenitors during postnatal bone development. RUNX2 stimulates the differentiation of MSCs during intramembranous ossification, leading to the formation of anterior osteoblasts and immature osteoblasts. Furthermore, Liu and Lee [30] analyzed the regulatory network that controls bone formation, taking into account several transcription factors such as Msh homeobox 2, TWIST, and promyelocytic leukemia zinc-finger protein. RUNX2 promotes trans-differentiation of terminal hypertrophic cartilage during endochondral ossification [31]. While our knowledge of the transcriptional mechanisms underlying osteogenesis is expanding rapidly, many questions remain about how RUNX2 contributes to skeletal development and how osteogenesis is orchestrated. RUNX2 is essential for breast growth because it controls the expression of a number of important mammary gland genes [32]. Since RUNX2 is expressed in both the basal and luminal cells of the mammary gland, it is likely involved in the development of the gland. Osteopontin was shown to be a potential RUNX2 target and to have a role in breast development in breast epithelial cells [33,34]

## 4. Deregulation of RUNX2 Signaling in Osteogenesis May Lead to Oncogenesis

RUNX2 controls osteoblast lineage specification, osteoblast development, osteoblast maturation, and osteoblast terminal differentiation via a network of interconnected pathways. RUNX2’s direct links to widely active developmental pathways suggest that pluripotent MSCs may generate primitive osteoblast progenitor cells. In order to promote a pre-osteoblastic phenotype, canonical Wnt factors and members of the Hh family are known to inhibit MSCs’ differentiation into adipocytes and chondrocytes [35]. Several recent investigations have shown a connection between RUNX2 and the canonical Wnt pathway, which regulates osteoblast commitment. The beta-catenin and RUNX2-dependent activation of the fibroblast growth factor 18 (FGF18) gene in mesenchymal stem cells (MSCs) has been linked to the suppression of chondrogenesis and the stimulation of osteogenesis [36].

The canonical Wnt pathway plays a crucial role in osteoblast differentiation. Wnt signaling is required to further commit osteoprogenitors to the osteoblast lineage [37], following the RUNX2-mediated transcriptional activation of the osterix (Osx1/SP7) gene [38]. Via its interaction with SMAD proteins induced by BMP and TGF, RUNX2 promotes differentiation shortly after lineage commitment. By increasing osterix expression [38,39] in RUNX2-independent osteoprogenitors, BMP2 promotes osteoblast formation. However, the formation of the RUNX2–SMAD complex is crucial for BMP/TGF signaling to reach its conclusion since it stimulates the transcription of late osteoblast markers [40].

Dedicated osteoblasts divide and move to new locations before becoming dormant and fully differentiating. Canonical Wnt signaling via LRP5 [41] and indirectly via Src/ERK and PI3K/Akt [42,43] promotes osteoblast proliferation and survival. During the G1 phase, when osteoblasts are differentiating and expanding, the protein level of RUNX2, a transcription factor known to limit osteoblast proliferation, is highest. RUNX2 activity is normally downregulated throughout the G1-to-S transition and the succeeding S, G2, and M phases, but if quiescence is induced, it stays high throughout the G0 period [21]. During mitosis, RUNX2 is found in its resting state in the active nucleolar organizing areas, where it maintains its repression of the genes that code for ribosomal RNA [44]. The epigenetic control of protein-encoding genes during mitosis, often known as “bookmarking” [45,46], may involve RUNX2 in some way. In the absence of RUNX2, higher growth potential is created in vitro, but an increase in RUNX2 causes cell cycle exit when either contact inhibition or serum deprivation is applied [8]. The osteosarcoma cell line SAOS-2 is able to upregulate BAX expression and induce apoptosis thanks to RUNX2 being triggered by BMP/SMAD signaling [47].

RUNX2 not only has a favorable function in controlling cell development but also promotes cell proliferation and survival. Maximal RUNX2 levels are required for cell division to proceed during the G1 phase [8]. Increased sensitivity to mitogenic signaling via the cyclic AMP and G-protein-coupled receptor signaling pathways [48] is a result of the RUNX2 repression of the transcription of p21/CDKN1A/WAF1/CIP1, which encodes a cyclin-dependent kinase inhibitor that arrests cells in G1. The nitric oxide (NO) treatment of the MG-63 osteosarcoma cell line was shown to enhance RUNX2-mediated BCL2 expression, which improves cell viability under oxidative stress [49], in contrast to results that suggested that RUNX2 upregulates BAX expression in the SAOS-2 cell line [8]. Matrix metalloproteinase (MMP) gene transcription for MMP13 is upregulated in response to NO signaling via cyclic guanosine 3′,5′ -monophosphate (cGMP) [50,51]. A possible explanation is that RUNX2 is phosphorylated in a critical spot by protein kinase G (PKG).

Platelet-derived growth factor (PDGF), transforming growth factor (TGF), and insulin-like growth factor (IGF) gradients are involved in chemotactic osteoblast migration during bone formation and remodeling [52,53,54]. The upregulation of RUNX2 by PI3K-Akt promotes this migration [55]. In the last stages of differentiation, the cell cycle exits, and all osteoblast phenotypic markers are expressed. By inhibiting S-phase cyclin-dependent kinases, RUNX2 enables cell cycle exit via the induction of pRB dephosphorylation [56]. Cell cycle exit at this stage requires active, hypophosphorylated pRB [57], which is bound by RUNX2 and the transcription factor HES1 [58]. Osteocalcin is one of many late indicators of osteoblast development for which transcription is coactivated by the RUNX2–pRB complex [59]. Histone acetyltransferases (HATs) p300 and p300/cyclic AMP receptor element-binding protein binding protein-associated factor (PCAF) [60,61] are recruited to activate osteocalcin with the monocytic leukemia zinc finger protein (MOZ) and the MORF [60,62]. In addition to alkaline phosphatase (AP), osteopontin (OP), bone sialoprotein (BSP), and collagen type I (COL-1), a variety of additional indicators have been discovered [40].

RUNX2 interacts with a broad number of corepressor proteins depending on its phosphorylation and developmental stage. By inhibiting p21/CDKN1A/WAF1/CIP1 and osteocalcin, histone deacetylases (HDACs) 6 and 3 interact with RUNX2 to control osteoblast formation during proliferation and terminal differentiation [48,61]. RUNX2 and other histone deacetylase (HDAC) proteins form complexes with the corepressors mSin3a, TLE/Groucho, and Yes-associated protein (YAP) to repress the expression of osteoblast-specific genes like osteocalcin [6,63]. HDAC4 binds RUNX2 and suppresses its inherent DNA-binding activity [64], thereby inducing transcriptional repression. Studies are presently undertaken to define the intricate relationship between RUNX2 and the downstream components that regulate osteoblast development. This is important because the proteins with which RUNX2 forms multi-subunit complexes have a large influence on its transcriptional regulation and tissue-specific nature.

## 5. RUNX2 May Be an Important Player in Osteosarcoma

Once MSCs decide to become osteoblasts, RUNX2 levels rise progressively throughout normal bone growth, reaching a peak in early osteoblasts. Over the last several years, many investigations on osteosarcoma samples have shown abnormally high levels of RUNX2 protein. The results of the few clinical studies that have looked at RUNX2 have been striking. The first study we could locate documenting RUNX2 immunoreactivity in osteosarcomas was written by Andela et al. [65], who examined 11 malignant pathologic specimens and reported RUNX2 immunopositivity in all of them.

RUNX2 expression was linked to clinical progression indicators in patients with osteosarcoma in three more investigations published in the last year. Comparing 22 osteosarcomas, we found that the average mRNA overexpression of RUNX2 was 3.3-fold greater in tumors that had a poor response to neoadjuvant chemotherapy (90% necrosis) than in tumors that had a favorable response (>90% necrosis) [66]. It has been established that RUNX2 mRNA expression is greater in cancer samples than in normal human osteoblasts [67,68]. Won and co-workers also found that, while 23 percent of cores (11/48) showed significant quantities of RUNX2, 60 percent of cores (29/48) showed low levels. High RUNX2 expression was associated with greater metastatic risk and lower overall survival [69]. Another study compared the immunoreactivity of RUNX2 in various patient samples and found that 65% (12/20) of biopsy samples and 73% (8/11) of metastatic tumors were positively stained. Only 16% (4/25) of resections performed after chemotherapy showed RUNX2 staining, according to the same research. These studies provide preliminary evidence that RUNX2 may be useful for making prognoses [70]. Due to RUNX2’s complex activity in growing osteoblasts, it is possible that its dysregulation contributes to osteosarcoma pathogenesis. Regulating RUNX2 is independent of cell cycle progression in this particular cell line.

SAOS-2 and the protein stay at relatively high levels throughout the rest of the cell cycle despite being downregulated during the G1-to-S transition [71]. RUNX2 interacts selectively with hypophosphorylated pRB [72] near the conclusion of osteoblast development, heralding the start of cell cycle withdrawal. Osteosarcoma is one of the few types of cancer in which pRB inactivation is widespread, occurring in 50–70% of tumors [68,69,71,72,73]. The inability of cells to exit the cell cycle in response to the RUNX2–pRB signal would allow for the unrestricted proliferation of osteoprogenitor cells and may increase genomic instability [74].

The p53–MDM2 pathway is also related to pRB [75], and this association may explain why RUNX2 is important for bone health. Patients with Li–Fraumeni syndrome and their relatives are at an increased risk for developing osteosarcoma due to abnormalities in the p53 pathway. The loss of p53 is more important than the deletion of pRB in mice models of osteosarcoma susceptibility [76]. The genetic perturbation of the p53–MDM2 pathway blocks RUNX2-dependent osteoblastic development, and the loss of p53 function increases RUNX2 accumulation during differentiation [77]. When p53 is deleted, RUNX2 protein levels rise in numerous growth-factor-independent osteosarcoma cell lines [68,69,71,72,73,74,75,76,77], in contrast to the low levels seen in primordial or immortalized osteoblasts. Samples from patients with osteosarcoma have been shown to have amplified copies of the 6p12-6p21 genes [68,69,71,72,73], which may be caused by or at least contribute to the loss of p53 activity in these tumors.

RUNX2 promotes cell cycle exit and terminal differentiation by interacting with the cyclin-dependent kinase inhibitor p27KIP1/CDKN1B [78]. RUNX2’s cell-cycle-dependent activity is regulated by phosphorylation by cyclin-dependent kinases (CDKs). Undifferentiated osteosarcoma is characterized by low levels of the p27KIP1 protein [56]. Despite not being published, our own aCGH investigation of 15 osteosarcoma patient samples revealed CDKN1B loss in 9 of the 15 samples. Because of this, RUNX2 signaling relies on the tumor suppressors pRB and p27KIP1 to successfully regulate proliferation and induce osteoblast maturation in the absence of these factors. The loss of p53, the primary trans-activator of this cyclin-dependent kinase inhibitor, and increased RUNX2 protein levels (which transcriptionally block the p21CIP1/CDKN1A gene) are two potential causes of decreased p21CIP1/CDKN1A expression [79,80]. Osteosarcoma chemotherapy and radiation experiments demonstrate that inhibiting p21CIP1 prevents cells from entering growth arrest and from repairing DNA after damage.

The inadequate regulation of RUNX2’s pro-differentiation and tumor suppressor activities may prevent MSCs predisposed to the osteoblast lineage from maturing into fully functional osteoblasts. Recently, it was shown that Notch1 indirectly downregulates RUNX2 by increasing the expression of the cyclin D1-dependent kinase CDK4, which ubiquitinates RUNX2. However, no functional investigations have yet connected the inactivation of RUNX2 directly to osteosarcoma metastasis, despite the fact that there is a connection between increased Notch signaling and lung metastatic capabilities in osteosarcoma cell lines [81].

Despite its conventional role as a tumor suppressor, recent data suggests that RUNX2 may have a direct involvement in promoting neoplasia, especially in prostate and breast malignancies [82]. Osteosarcomas and other malignancies may stimulate a variety of signaling pathways, including PI3K/Akt, Wnt, BMP/TGF, MAPK/ERK, and Notch signaling [83,84,85,86]. Overexpressing RUNX2 in implanted prostate cancer cells [5] activated genes required for osteolytic illness, such as PTH-related protein (PTHrP) [87] and interleukin 8 (IL8) [88]. IL8 increases osteolysis by increasing osteoclast production independently of RANKL [89], in contrast to PTHrP and RUNX2, which suppress the expression of RANKL but enhance osteoclast formation and bone resorption. However, RANK/RANKL is overexpressed in certain osteosarcomas [90,91], a kind of cancer that combines osteolytic and osteoblastic processes. Researchers Akech et al. [5] found that RUNX2 overexpression in prostate cancer activated genes involved in invasion, metastasis, and angiogenesis [5]. These results add to the growing body of data suggesting that RUNX2 promotes metastasis in prostate cancer cell lines [92] and in metastatic patient specimens [93]. These findings corroborate the findings of other research [94,95] that showed that RUNX2 expression is necessary for osteolytic illness resulting from breast cancer metastasis.

RUNX2 may act as a tumor suppressor or an oncoprotein, depending on its expression, its environment, and its regulation. The oncogene overexpressions of RUNX2 and MYC generate proteins that aid in cancer cell maintenance and proliferation [96]. Since galectin-3 (LGALS3) is expressed in pituitary tumors [97], it is possible that its expression is also elevated in osteosarcomas [98]. However, owing to our limited understanding of the repercussions of its dysregulation in osteoblasts, the involvement of this protein in bone tumorigenesis is still unknown. The bone-remodeling hormone parathyroid hormone (PTH) induces apoptosis in osteoblasts, although high levels of RUNX2 suppress this process [99]. The first mouse model of osteosarcoma was created by overexpressing the proto-oncogene FOS [100]. In order to promote metastasis, the FOS protein product interacts with RUNX2 to boost the AP-1-mediated transcription of the MMP13 gene [101,102]. The role of RUNX2 in breast-cancer-mediated bone metastasis is depicted in Figure 2.

## 6. Conclusions

It is crucial for healthy bone formation that the different actions of RUNX2 during osteoblast development are properly regulated. Researchers have shown that committed osteoprogenitor cells are unable to mature into osteocytes due to cell cycle dysregulation, which is reminiscent of the behavior of certain types of osteosarcoma. The emergence of well-characterized osteosarcoma cell lines and the development of mice models of the illness demonstrate that osteosarcomas, like other malignancies, present a spectrum of differentiation statuses. It has been established that elevated RUNX2 expression in osteoblast progenitor cells interferes with the normal development and cell cycle control of osteoblasts. The carcinogenic potential of RUNX2 overexpression and the possibility that its tumor suppressor functions are dysregulated suggest that the gain and amplification of chromosome 6p12-p21 plays a role in the etiology of osteosarcoma. Based on our unpublished research, we know that the genetic amplification of this protein’s DNA and the interruption of its degradation contribute to its overexpression. The increase and amplification of RUNX2 due to chromosome 6p12-p21 instability has been shown by several investigations, including biopsies, to be an early event in the etiology of osteosarcoma. For a long time, osteosarcoma’s inherent complexity has stood in the way of researchers’ attempts to comprehend the illness, locate prognostic or predictive indicators, and create curative therapies. Although our lab’s results suggest RUNX2 may be predictive of responses to regular chemotherapy, further research is required to determine its cancer-specific role. RUNX2 expression may be correlated with the efficacy of therapy for osteosarcoma tumors; however, more patient studies are required to draw any firm conclusions. In conclusion, RUNX2 offers promise as a prognostic factor and therapeutic target based on its well-established activities and its frequent increase in the tissues of patients with osteosarcoma.

## Figures and Tables

**Figure 1 cancers-15-03247-f001:**
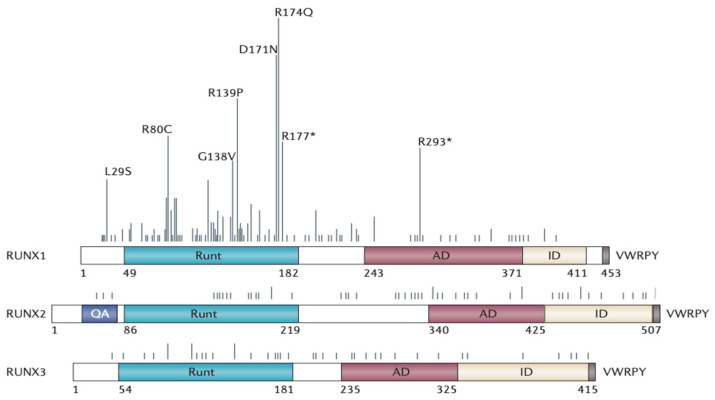
A breakdown of where somatic mutations are located in each of the three human RUNX genes. The Runt DNA-binding domain, the activation domain (AD), the inhibitory domain (ID), and the VWRPY motif at the extreme carboxyl terminus are shown. These domains are responsible for interacting with the co-repressor Groucho (also known as TLE1). The relative frequency of different amino acid changes in human cancer is shown by the bars located above each figure, and an asterisk denotes the presence of nonsense replacement. RUNX2 is the only gene known to have tandem repetitions of the amino acids glutamine and alanine, and these repeats are referred to as QA. The Catalog of Somatic Mutations in Cancer was used to compile the data for this study. When it comes to RUNX1, the majority of mutations are found in the Runt domain. These mutations often result in a loss of the protein’s capacity to bind to DNA. In the C-terminal region, mutations are uncommon, but when they do occur, they are linked to sporadic myelodysplastic syndrome and acute myeloid leukemia (AML). It is possible that mutations in the C-terminal regions will also lead to an increased incidence or risk of AML transformation193. It is not known if cancer caused by point mutations in RUNX2 has any functional impact. Cancers of the bladder and the stomach have been related to variations in the amino acids found in the RUNX3 gene’s Runt domain. (Image obtained with permission [11]).

**Figure 2 cancers-15-03247-f002:**
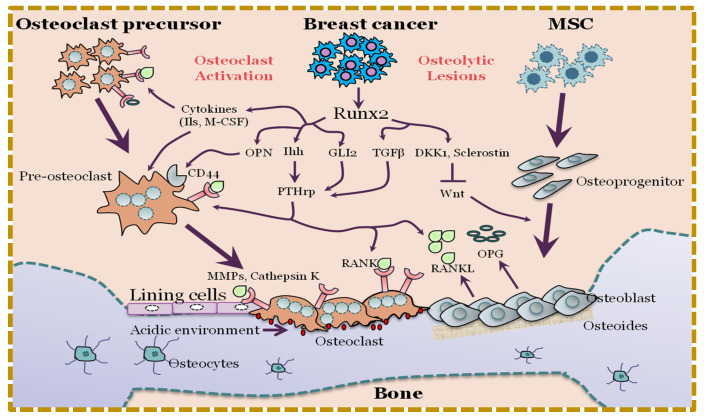
Role of RUNX2 in breast-cancer-mediated bone metastasis. (Image obtained with permission [103]).

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
