# Peer review of "RUNX Family as a Promising Biomarker and a Therapeutic Target in Bone Cancers: A Review on Its Molecular Mechanism(s) behind Tumorigenesis"

_cancers, 2023, doi:10.3390/cancers15123247_

Round 1

Reviewer 1 Report

The manuscript provides a comprehensive and insightful analysis of the role of the RUNX family of transcription factors in tumorigenesis. The author's ability to clearly explain the complex molecular mechanisms involved is commendable, making the content accessible to a wide range of readers. The manuscript highlights the significance of understanding the precise processes underlying the occurrence and prognosis of various malignant tumors, with a particular focus on bone cancer. The inclusion of recent research findings and the discussion of potential therapeutic implications make this manuscript a valuable resource for researchers and clinicians interested in cancer biology and targeted therapies. Overall, this manuscript is well-written, informative, and contributes to our understanding of the tumorigenic role of the RUNX family in important cancers. Based on a thorough review of the manuscript titled “RUNX Family as a Promising Biomarker and a Therapeutic Target in Bone Cancers: A Review on its Molecular Mechanism(s) Behind Tumorigenesis”, I would like to recommend its acceptance for publication in Cancers-MDPI. The manuscript provides a comprehensive analysis of the role of the RUNX family of transcription factors in tumorigenesis, particularly in relation to bone cancer.

The author has demonstrated a strong understanding of the subject matter and has effectively conveyed complex molecular mechanisms in a clear and accessible manner. The manuscript incorporates up-to-date research findings, offering valuable insights into the precise processes underlying the occurrence and prognosis of various malignant tumors.

The manuscript's contribution to the field is significant, as it highlights the emerging role of the RUNX family as a therapeutic target for bone cancer. The discussion of potential clinical implications and the exploration of molecular mechanisms provide valuable information for both researchers and clinicians.

The writing style is engaging, and the organization of the manuscript is logical, allowing readers to follow the arguments and conclusions easily. The inclusion of appropriate references and supporting evidence further strengthens the manuscript's credibility and scientific rigor.

In summary, this manuscript makes a valuable contribution to the understanding of the tumorigenic role of the RUNX family in important cancers. It is well-written, informative, and presents a comprehensive analysis of the subject matter. Therefore, I recommend accepting this manuscript for publication in “Cancers-MDPI”.

Please note that minor revisions may be required to address the title and it may be suggested to change it as “RUNX2 as a Promising Biomarker and a Therapeutic Target in Bone Cancers: A Review on its Molecular Mechanism(s) Behind Tumorigenesis

Author Response

Dear reviewer, we express our sincere gratitude for highlighting our manuscript in a positive way.  Regarding your suggestion, kindly permit us to keep the same title as we have dealt with the whole Runx family in the manuscript and from an author's and a reader's point of view, we feel the current title is appropriate for the manuscript. Thank you once again for the comments and suggestions. 

Reviewer 2 Report

In this article, Vimalraj and Sekaran conducted a review of the transcription factor runt-related protein (RUNX) family as biomarkers and therapeutic targets in bone cancer, with particular focus the underlying molecular mechanisms. They highlighted that RUNX family of transcription factors plays an important role in bone development by the formation of osteoblasts from bone marrow mesenchymal stem cells, in tumour progression and in particular osteosarcoma via its regulation of many upstream signalling pathways and downs stream molecules.  They examined the various potential underlying mechanisms responsible for the tumorigenic role of RUNX in various cancers. They concluded that RUNX has potential as a novel therapeutic target in cancer, and therefore its in-depth study of related regulatory processes could help to the creation of newer medications with improved clinical efficacy.

Overall, this is an important area of research and I have the following comments/suggestions which should be checked:

 1)  Lines 64-66. The authors mention that: "As an example, our lab has recently shown (8) that RUNX2 activation of MMP1 significantly corresponds with a poor prognosis by promoting the aggressiveness and chemoresistance of TNBC cells".  However, reference 8 is an study which has been done by colleagues in a lab in Canada not India. Please check.

 2) Lines 126-127. Authors included figure 1, which is identical to the figure in reference 106 which was published in 2015. I am not sure why an identical figure another publication was used here. It would have been better to have an updated version of that figure by adding any new information since 2015. If not, there is no need to include this figure and it can be referred too.

 3) Lines 129-132. References should have been cited in the numerical order 19. Also I was not able to see citation to reference 20 in the main text and refs 22-25 were cited before ref 20 and 21.  

 4) Lines 235-238. Authors stated that "Comparing 22 osteo-sarcomas,, we found that the average mRNA overexpression of RUNX2 was 3.3-fold greater in tumours that had a poor response to neoadjuvant chemotherapy (90% necrosis) than in tumours that had a favourable response (>90% necrosis)”. Where have they reported this and where is the reference for this study?  

5) Lines 243-26.  “Another research compared the immunoreactivity of RUNX2 in various patient samples and found that 65% (12/20) of biopsy samples and 73% (8/11) of metastatic tumours were positively stained. Only 16% (4/25) of resections performed after chemotherapy showed RUNX2 staining, according to the same research”. Which study they are referring to and where is the reference for that study?

6) Again Figure 2 is also identical to a Figure from reference 107. There is no need to copy the same figure from another publication in 2017, unless there are more information which can be added by updating that figure.

Author Response

 1)  Lines 64-66. The authors mention that: "As an example, our lab has recently shown (8) that RUNX2 activation of MMP1 significantly corresponds with a poor prognosis by promoting the aggressiveness and chemoresistance of TNBC cells".  However, reference 8 is an study which has been done by colleagues in a lab in Canada not India. Please check.

Response: We extremely apologize for the error. We have corrected the revised manuscript as “As an example, a recent study has shown [8] that…”

 2) Lines 126-127. Authors included figure 1, which is identical to the figure in reference 106 which was published in 2015. I am not sure why an identical figure another publication was used here. It would have been better to have an updated version of that figure by adding any new information since 2015. If not, there is no need to include this figure and it can be referred too.

Response: We completely agree with the reviewer that the figure is completely indistinguishable, as stated in their comments. We acknowledge that the image has not been modernized; however, from the perspective of the reader, it will be beneficial for them to be able to easily refer to the image if it is included in the section that has been conserved. This will make it much simpler for them to comprehend the material that is being discussed. As a result, we would like to ask the reviewer if it would be possible for us to continue using the figure in the current manuscript.     

 3) Lines 129-132. References should have been cited in the numerical order 19. Also I was not able to see citation to reference 20 in the main text and refs 22-25 were cited before ref 20 and 21.  

Response: We are sorry for the mistake. We have corrected the revised manuscript. Thanks to the reviewer.

 4) Lines 235-238. Authors stated that "Comparing 22 osteo-sarcomas,, we found that the average mRNA overexpression of RUNX2 was 3.3-fold greater in tumours that had a poor response to neoadjuvant chemotherapy (90% necrosis) than in tumours that had a favourable response (>90% necrosis)”. Where have they reported this and where is the reference for this study?  

 Response: Sorry. We have added appropriate references.

5) Lines 243-26.  “Another research compared the immunoreactivity of RUNX2 in various patient samples and found that 65% (12/20) of biopsy samples and 73% (8/11) of metastatic tumours were positively stained. Only 16% (4/25) of resections performed after chemotherapy showed RUNX2 staining, according to the same research”. Which study they are referring to and where is the reference for that study?

 Response: Sorry. We have added appropriate references.

6) Again Figure 2 is also identical to a Figure from reference 107. There is no need to copy the same figure from another publication in 2017, unless there are more information which can be added by updating that figure.

Response: Kindly accept the same response given for comment 2. Response: We completely agree with the reviewer that the figure is completely indistinguishable, as stated in their comments. Having said that, the figure was drawn by me for one of my colleagues in the lab. In recognition of this fact, my name appears in the manuscript acknowledgement section that was referred to. In addition, with regard to the information, we acknowledge that the image has not been modernized; however, from the perspective of the reader, it will be beneficial for them to be able to easily refer to the image if it is included in the section that has been conserved. This will make it much simpler for them to comprehend the material that is being discussed. As a result, we would like to ask the reviewer if it would be possible for us to continue using the figure in the current manuscript.     

I wanted to take a moment to express my sincere gratitude for reviewing our manuscript and providing invaluable feedback. Your expertise and attention to detail have greatly enhanced the quality of my work, and I am truly grateful for your time and effort.

Your thoughtful comments and constructive criticism have helped me to refine my ideas and improve the overall presentation. Your thoroughness in reviewing every aspect of the [project/article/document] has been instrumental in making it more comprehensive and impactful.

Once again, thank you for your invaluable contribution. I am truly fortunate to have received your guidance, and I am confident that your insights will make a significant difference in the final outcome.
